# Panoptic View of Prognostic Models for Personalized Breast Cancer Management

**DOI:** 10.3390/cancers11091325

**Published:** 2019-09-07

**Authors:** Geetanjali Saini, Karuna Mittal, Padmashree Rida, Emiel A. M. Janssen, Keerthi Gogineni, Ritu Aneja

**Affiliations:** 1Department of Biology, Georgia State University, Atlanta, GA 30303, USA; 2Department of Pathology, Stavanger University Hospital, 4011 Stavanger, Norway; 3Department of Hematology and Medical Oncology, Emory University School of Medicine; Atlanta, GA 30322, USA

**Keywords:** breast cancer, prognosis, prediction, liquid biopsy, immunohistochemistry, digital pathology, multigene assays

## Abstract

The efforts to personalize treatment for patients with breast cancer have led to a focus on the deeper characterization of genotypic and phenotypic heterogeneity among breast cancers. Traditional pathology utilizes microscopy to profile the morphologic features and organizational architecture of tumor tissue for predicting the course of disease, and is the first-line set of guiding tools for customizing treatment decision-making. Currently, clinicians use this information, combined with the disease stage, to predict patient prognosis to some extent. However, tumoral heterogeneity stubbornly persists among patient subgroups delineated by these clinicopathologic characteristics, as currently used methodologies in diagnostic pathology lack the capability to discern deeper genotypic and subtler phenotypic differences among individual patients. Recent advancements in molecular pathology, however, are poised to change this by joining forces with multiple-omics technologies (genomics, transcriptomics, epigenomics, proteomics, and metabolomics) that provide a wealth of data about the precise molecular complement of each patient’s tumor. In addition, these technologies inform the drivers of disease aggressiveness, the determinants of therapeutic response, and new treatment targets in the individual patient. The tumor architecture information can be integrated with the knowledge of the detailed mutational, transcriptional, and proteomic phenotypes of cancer cells within individual tumors to derive a new level of biologic insight that enables powerful, data-driven patient stratification and customization of treatment for each patient, at each stage of the disease. This review summarizes the prognostic and predictive insights provided by commercially available gene expression-based tests and other multivariate or clinical -omics-based prognostic/predictive models currently under development, and proposes a more inclusive multiplatform approach to tackling the challenging heterogeneity of breast cancer to individualize its management. “The future is already here—it’s just not very evenly distributed.”-William Ford Gibson

## 1. Introduction

It has become increasingly apparent over the past few decades that in order to grasp and effectively combat the heterogeneity that typifies breast cancer (BC), high-granularity tumor biomarker profiling is not merely desirable, but in fact, indispensable. Increasingly affordable novel technologies and deep-content analytics are enabling molecular profiling of tumors throughout the course of patient care. Specifically, the interrogation of the tumor tissue as well as its genome, transcriptome, epigenome, proteome, metabolome, and other aspects of both the tumor and the host characterize the burgeoning field of -omics research, which uses bioinformatics and computational technology to explore the mechanistic properties of molecules. The discovery and validation of definitive genetic and phenotypic biomarkers have emerged as the cornerstone of predictive and prognostic testing that can be used to parse patients with BC into subgroups and risk categories for analysis, thus helping to identify targeted treatments for each patient’s unique molecular profile by “matching the pill to the ill.”

Molecular diagnostic tests, however, are only as good as the biomarkers they identify or measure. Like any diagnostic test, a tumor biomarker test, whether genetic or phenotypic, must have analytic and clinical validity as well as clinical utility. In the current BC landscape, several commercial genomic assays available to oncologists are transforming patient treatment. The assays to determine the status of the biomarkers such as estrogen receptor (ER), progesterone receptor (PR), and human epidermal growth factor receptor 2 (HER2) have long been used for the primary identification of tumors for targeted therapy and prognostication. Other assays serve as companion diagnostic (CDx) tests due to their ability to predict tumor response to specific cancer therapy drugs. These CDx assays are useful for existing as well as new drugs, increasing the clinical value of therapy by selecting for potential responders or excluding patients at risk for severe adverse effects. For pharmaceutical companies, CDx assays facilitate regulatory approval of new therapeutic regimens, enhance the probability of success in clinical trials, and in combination with adaptive trial designs, make trials more cost-effective.

Clinical decision making in BC chiefly relies on determining clinicopathologic features of a tumor as well as on protein- and gene-based biomarker panels. The review, ordered in a broadly chronological fashion, begins with examining these mainstays and developments therein. This is followed by a discussion on more recent technologies for data gleaning (liquid biopsies, tumor microenvironment-based markers and metabolomics) that are receiving tremendous attention in consonance with the significant information they generate. The advances in digital pathology, particularly the use of convoluted and deep neural networks, adding invaluable tools to the repertoire, have also been covered in the review. Figure 1 captures a comprehensive picture of the current and emergent approaches in clinical management of BC patients, which are discussed in the subsequent sections.

## 2. Integrating Clinico-Pathological Variables to Frame Breast Cancer Prognostic Models

The clinico-pathological parameters, typically the tumor size, grade, nodal status, age of patient are crucial variables that when combined meaningfully can effectively determine the prognosis in BC patients. In primary breast cancer, the Nottingham prognostic index (NPI) calculated using the tumor size, lymph-node stage, and pathological grade is considered the standard [1]. The Nottingham prognostic index plus (NPI+) is an improvement on this and is based on an initial determination of the biological class of the tumor combined with clinicopathologic prognostic variables. The NPI+ can predict the risk of metastases and is touted to provide enhanced risk stratification as well as predict long-term survival [2]. Kwon et al. have proposed a modified Nottingham prognostic index (MNPI) for stratifying patients with stage I to III of triple negative BC (TNBC), a notoriously aggressive type of BC. The index incorporates information on the tumor size, LN status, and tumor grade according to a modified Scarff-Bloom-Richardson (MSBR) grade, and is potentially an important prognostic tool for patients with TNBC [3]. Given the highly heterogeneous nature of this BC subtype, and the existence of distinct molecular subtypes within TNBC, it is possible that prognostic models assigning appropriate weights to subtype-specific variables may need to be developed for the individual molecular subtypes of TNBC. The Van Nuys prognostic index (VNPI) is a scoring system (based on the tumor size, margin width, grade, comedonecrosis and age) that assists the treatment decision making in ductal carcinoma in situ patients (DCIS) [4]. In addition to these, several web-based prognostic tools derived from clinicopathologic variables, e.g., Adjuvant! Online, PREDICT, Clinical Treatment Score post–5 years (CTS5) are freely available and may provide adjunctive information in clinical decision making [5,6]. The clinicopathologic variables in harness with the gene and protein-based biomarkers can provide far superior and robust prognostication.

## 3. Immunohistochemistry-Based Prognostic Assays for Breast Cancer

The most widely used protein markers (ER, PR, HER2, Ki67) are well-described predictive markers for hormonal and anti-HER2 therapy. Immunohistochemistry (IHC) is a commonly used technique to measure the expression of these biomarkers. Additionally, in situ hybridization (e.g., fluorescence in situ hybridization and chromogenic in situ hybridization) is performed to quantify HER2 gene amplification which often results in HER2 overexpression. The IHC4 index is a non-commercial algorithm that assesses these four protein markers, generating a disease recurrence score. A lack of validation studies and poor reproducibility has marred this prognostic tool’s prospects in general clinical application [7,8]. The IHC4 and CTS (a clinical treatment score based on clinico-pathological parameters), have been combined by Cuzick et al., to yield an overall prognostic score that may prove useful in predicting risk of recurrence in ER positive BC patients [7,9].

The urokinase plasminogen activator (uPA) and its inhibitor protein plasminogen activator inhibitor-1 (PAI-1) have shown to be promising independent prognostic markers and have attained the highest level of evidence (LOE-1a) in terms of clinical utility in BC [10,11]. In several European countries, the value of uPA and PAI-1 as biomarkers for a predictive outcome in LN-negative BC has been validated in both retrospective and prospective studies [11,12]. The high levels of these markers (measured by an American Society of Clinical Oncology [ASCO] recommended enzyme-linked immunosorbent assay [ELISA] based assay, using extracts of fresh or freshly frozen breast tumor tissue) correlate strongly with an adverse prognosis and increased benefits from adjuvant chemotherapy [10,13,14]. Compared with Oncotype Dx and MammaPrint, uPA/PAI-1 assessment is more convenient, cost-effective and may even provide greater prognostic and predictive value for BC outcomes [15,16,17]. Unfortunately, no validated assay for uPA/PAI-1 is currently available.

The multiplexed immunohistochemistry (mIHC) is a recent tool that allows for simultaneous probing of several protein biomarkers on the same biological sample. The staining of samples can be chromogenic or fluorescent. This is crucial when the tumor sample size is limiting. The mIHC can provide valuable insights about the co-expression and spatial distribution of many targets without compromising tissue integrity [18]. Despite the fact that IHC-based assays do not require tissue microdissection and are more clinically facile, pre-analytic tissue processing can significantly impact test results [19]. Other studies have uncovered additional concerns with IHC-based assays, such as inconsistent performance of IHC reagents, antibodies and widespread variation in slide scoring, which can significantly influence test results and call into question the IHC platform’s consistency and reliability [19].

## 4. Gene-Centered Biomarkers: Translating Molecular Complexity of Tumors by Gene Expression-Based Assays

Many currently available prognostic and predictive tests in breast oncology utilize genomic and gene expression-based biomarkers, a trend that was spurred by the continually decreasing costs of whole-genome, gene panel, and RNA sequencing, and that requires only small sample volumes for processing. Multigene assays often interrogate one or more pathways that drive tumor biology, and predict the natural progression of the disease (with or without therapeutic intervention) based on its inherent aggressiveness. Thus, these tests help distinguish patients who need more aggressive treatments from those for whom the current standard-of-care may suffice, reducing healthcare expenditures. Several multigene tests (MGTs) are now routinely used in the clinical setting, and have been described in previous literature [20,21,22,23,24,25,26,27,28,29,30,31,32,33,34,35,36,37,38,39,40,41,42,43,44,45,46,47,48,49,50,51,52,53,54,55,56,57,58,59,60,61,62,63,64,65,66,67,68,69,70,71]. The focus of this review is on more recent and emerging technologies, and these MGTs are summarized in Table 1 and their gene/protein signatures illustrated in Figure 2.

The performance of six multigene signatures were compared in women with early ER-positive BC who underwent endocrine therapy for 5 years. These multigene signatures included the oncotype Dx recurrence score, the PAM50-based Prosigna risk of recurrence (ROR), the breast cancer index (BCI), EndoPredict (EPclin), the clinical treatment score, and the 4-marker immunohistochemical score. The BCI, EPclin, and the PAM50-based Prosigna ROR showed significant prognostic value for predicting the overall and late distant recurrence in patients with lymph node (LN)-negative disease. The results from this telling study have proven useful for guiding oncologists and patients in choosing the most suitable testing to inform decisions regarding chemotherapy and/or extended endocrine therapy. These tests, however, have provided limited prognostic information for patients with node-positive disease [72].

Identifying specific mutations such as in the *BRCA1*, *BRCA2*, *PALB2* and *PTEN* genes by sequencing, can be especially useful in selecting BC patients who may be eligible for poly ADP-ribose polymerase (PARP) inhibitors [73]. Comprehensive BRCA testing is offered by multiple companies to identify BC patients with germline BRCA mutations. The myCHoice HRD (Myriad Genetics) CDx is a next-generation sequencing homologous recombination deficiency (HRD) assay that assesses both *BRCA1* and *BRCA2* as well as tumoral genomic instability. The assay uses DNA extracted from FFPE or frozen tumor tissue and labels a tumor as homologous recombination-deficient or -nondeficient, thus identifying patients who are most likely to benefit from treatment with PARP inhibitors. The BRACAnalysis CDx (Myriad Genetics) is an FDA-approved test for *BRCA1* and *BRCA2* for selecting patients suitable for olaparib treatment [74,75].

### 4.1. Number of Risk Categories: An Ongoing Debate

Of all aforementioned BC assays, only two prognostic assays, namely, Oncotype DX and Prosigna, originally assigned triple-category risk groups. However, Oncotype DX has recently discarded the intermediate recurrence score group. The intermediate recurrence score (RS) (initially 18 ≤ RS ≤ 30; re-classified as 11–25) in Oncotype DX had arguably been a grey area, leaving many women uncertain and concerned about their best treatment options (to omit chemotherapy or not) [29,33]. To address this issue, in 2006, began the TAILORx Study, one of the largest, randomized adjuvant BC treatment trials (in early stage, hormone-receptor–positive, HER2 negative, axillary node-negative BC patients). The intermediate group (RS 11–25) were randomly assigned to receive hormone therapy alone or hormone therapy plus adjuvant chemotherapy [27]. The results published in 2018 showed that women in this group did not additionally benefit from chemotherapy [76]. In the wake of these results, the intermediate RS group was eliminated and Oncotype DX now provides a binary stratification, that is, a low (0–25) or high (26–100) score with the former deriving no benefit from chemotherapy and the latter benefitting substantially from it. Thus, there is only one true triple-category risk group assay which is commercially available for breast cancer prognostication. The intermediate score category of the FDA-endorsed Prosigna PAM50-based MGT follows the ROR scoring system based on the LN spread of BC. If the LN is not affected (node-negative), the intermediate range is 41 ≤ ROR ≤ 60. However, if one or more (typically 1–3) LNs are affected, only a bimodal score is assigned. Several studies have shown that the node-negative ROR score is a better risk discriminator than the Oncotype DX RS [58].

### 4.2. Gene-Based Prognostic Assays: The Major Takeaways

In summary, MGTs offer several advantages. MGTs assign appropriate weightage to each variable and optimally extract information provided by multiple continuous variables. The information they provide is robust due to redundancy by capturing similar information from multiple genes. Even though these tests require specialized expertise to perform and interpret the results, overall, they prove to be cost-effective. Gene expression signatures are, however, unable to capture the prognostic information contributed by variables, such as tumor size or LN spread status, that lack an equivalent gene expression imprint. Therefore, there is a need to elevate the use of these clinicopathologic prognostic variables from merely providing complementary information to their incorporation as integral components of multivariate clinicogenomic risk models. These models could be built for distinct subtypes of BC because the prognostic value of each variable varies with the subtype and its unique tumor biology. In this regard, recently, Sparano et al., determined whether clinical risk assessment i.e., integrating tumor size and histologic grade, added prognostic and/or predictive information to the Oncotype DX RS. This was accomplished via secondary analyses of the TAILORx trial, and the results concluded that combining these information (from binary clinical-risk stratification and RS), provided prognostic information but was not predictive of chemotherapy benefit [77].

As the currently available MGTs assess the different pathways/gene sets, there remains the issue of discordance in risk assignment for a given patient sample by different MGTs. This ambiguity poses a challenge and increases the practical difficulty for clinicians in recommending a particular test, necessitating a head-on-head comparison of the MGTs. Although the prognostic gene signatures are being applied in clinical research trials, their ability to predict response to specific therapeutic agents is not as clear-cut. They are yet to graduate into routine clinical use for BC management. Furthermore, the gene expression-based tests focus only on the transcriptome, and it is becoming increasingly clear that gene expression and proteomic landscapes can diverge substantially. The MGTs entail gene expression profiling of a single sample from a tumor and evaluating a limited number of genes. They are thus, unable to adequately capture several aspects of the tumor’s phenotype, including intratumoral heterogeneity, the tumor microenvironment, and profiles of tumor infiltrating cells, all of which profoundly influence tumor biology and patient prognosis. Moreover, the current gene panels do not inform about driver mutations and epigenetic events involved in disease progression. Intratumoral heterogeneity has implications in therapy resistance as well as cancer progression and recurrence. Early detection of resistant subclones and tracking evolution of tumors requires longitudinal tissue sampling which is not practically feasible. This information can be gleaned by liquid biopsies instead, that offer a relatively non-invasive real-time monitoring method for serial sampling of circulating tumor DNA and tumor cells.

In many ways, nucleic acid markers seem ideal for elucidating disease pathology, and a fair number of molecular diagnostic tests exploit genetic variations (such as single-nucleotide polymorphisms [SNPs], mutations, and copy number variations [CNVs] that exist between abnormal and unaffected genomes. However, protein-based biomarkers more accurately profile the more relevant workhorses of the cell (functional proteins), and allow the capture of information pertaining to their subcellular localization, thus providing an edge over their nucleic acid counterparts. Furthermore, protein-based biomarkers are able to capture intratumoral heterogeneity and the expression of biomarkers in the tumor microenvironment, combining biological with morphological information. The following sections focus on the development of assays to analyse secreted protein molecules, circulating tumor cells and nucleic acids.

## 5. Liquid Biopsy Holds Promise for Guiding Breast Cancer Management

Liquid biopsy involves the analysis of circulating tumor cells (CTCs), cell-free circulating nucleic acids (circulating tumor DNA [ctDNA], and microRNA [miRNA]) and exosomes, released into the peripheral blood or urine from the primary tumor and/or metastatic deposits. It has recently emerged as a noninvasive prognostic, surveillance, and predictive tool in both early and metastatic BC that may complement, augment, or replace (in some cases) the use of tissue biopsy. The ease of sampling combined with the ability to monitor tumor burden or mutation changes temporally with ctDNA, allows for disease monitoring, the evaluation of therapeutic response, and molecular profiling in the advanced disease setting to determine therapeutic targets. The research in using ctDNA to tailor treatments has shown encouraging results. For example, the BELLE-2 trial and the SoFEA trial have demonstrated the clinical applicability of ctDNA to detect PI3K and ESR1 mutations, respectively, and the benefit of targeting these mutations with PI3K inhibitors and fulvestrant, respectively [78,79]. The NeoALTTO phase III trial found that the detection of ctDNA before starting NAC (neoadjuvant chemotherapy) correlated with lower pCR rates [80]. Thus, measuring ctDNA may aid in predicting the response to NAC. Another study in a series of patients with advanced breast cancer, showed that mutation levels in plasma samples (liquid biopsies) can provide similar information about clonal hierarchy as that determined by sequencing tissue biopsies. Thus, ctDNA can help characterize multifocal clonal evolution in metastatic cancers [81]. Furthermore, Chen et al., have used next generation sequencing (NGS) to analyze ctDNA and found a high specificity for predicting disease recurrence but a low sensitivity, possibly due to the low number of mutated DNA molecules in circulation [82].

In addition to ctDNA, CTCs may be used to predict the risk of relapse in early-stage disease [83]. The ECOG-ACRIN E5103 trial, in which samples were collected between 4.5 and 7.5 years after diagnosis, reported that patients who were positive for CTCs had a 21.7-fold higher chance of recurrence compared to those who were CTC-negative [83]. While CTC testing at a single time point is effective for risk stratification of patients, more studies are needed to define how these tests can be used in the clinical setting to make meaningful clinical decisions. As the circulation rate of CTCs is one per 1 × 10^9^ normal blood cells in metastatic cancer, their identification and isolation is particularly difficult [84]. Despite this, recent technologic advances have spawned new approaches for the selection and capture of CTCs, such as CTC-based assays that exhibit high specificity and low signal-to-noise ratio, especially in the detection of early-stage BC.

The CELLSEARCH Circulating Tumor Cell Kit (Menarini-Silicon Biosystems), a real time liquid biopsy currently licensed by Janssen Diagnostics among other companies, is the only CTC technology accredited by the FDA for the management of patients with metastatic BC. The semi-automated system uses a simple, actionable blood test based on an immunomagnetic enrichment technology [85]. A cut-off value of ≥5 CTCs per 7.5 mL of blood has been established, separating patients into shorter or longer survival groups [86,87,88]. Using the CELLSEARCH system, a few studies have validated CTC count as an independent prognostic factor for both metastatic and nonmetastatic BC, and found it to be more reproducible than radiology, detecting disease progression ahead by several weeks [88,89,90]. Monitoring CTC count during therapy potentially allows for the early detection of resistance to therapy. Due to its clinical validation and FDA approval, the assay has gained prominence as a type of gold standard within the field. However, the system uses expensive equipment and its cost is a major limitation. Furthermore, CTC capture is based on the presence of antigens on the cell surface that may result in both false-positive and false-negative results, with false-positives increasing during inflammatory conditions [91].

The EPISPOT assay (EPithelial Immuno SPOT) is also used to detect living CTCs although it employs a strategy different from CELLSEARCH [92,93]. A multicenter study observed an improved stratification of patients with metastatic BC (low- and high-risk groups) upon adding CTC status as measured by CELLSEARCH to that by EPISPOT, and found that a combination of both assays was the strongest predictor of OS (overall survival) [94].

The Nagrath Laboratory at the University of Michigan has created a superior CTC isolation technology in the form of a prototype wearable that continuously and directly traps CTCs from the patient’s blood. As the remaining blood products are returned post-CTC enrichment, larger blood volumes can be scanned, providing a lucid picture of tumor cell heterogeneity. This counters one of the major shortcomings in the current CTC technologies that rely on smaller blood draws, thus suffering from statistical variability [95].

When integrated at different points in the disease course, liquid biopsies can proffer information over and above that provided by standard clinicopathologic variables alone. Compared with CTC-based assays, ctDNA assays are superior when it comes to providing an individualized snapshot of a patient’s disease status, and they have greater sensitivity for early cancer detection. Unlike CTCs, ctDNA capture does not require specialized equipment [96]. Although ctDNA and CTC assays harbor immense potential, a lack of stringent studies and robust comparative assays are impairing their clinical application.

### 5.1. Serum Protein Markers

Videssa Breast (Provista Diagnostics) is a combinatorial multi-protein biomarker blood test for BC that evaluates 11 serum protein biomarkers (SPBs) and 33 tumor-associated autoantibodies (TAAbs) (Figure 2). The data are then combined with the patient age into a logistic regression algorithm, and the outcome is defined as a high protein signature (HPS) or low protein signature (LPS) [97]. One study validated the use of this noninvasive, actionable tool in detecting BC in women aged 50 years or younger with a low or intermediate risk who had abnormal or difficult-to-interpret imaging results (BI-RADS scores of 3 and 4). The Videssa Breast test used in conjunction with imaging results improved the diagnostic accuracy and reduced unnecessary biopsy by up to 67% when applied to cases that presented a challenging clinical assessment (compared with imaging alone), and the negative predictive value was 99%. Thus, in cases where mammogram results are abnormal, this tool can help clinicians identify the patients who are highly unlikely to have BC [98]. Another study demonstrated that the assay could reliably rule out BC in women with both dense and non-dense breasts [97].

A few FDA approved serum BC markers worth mentioning are the Carcinoma Antigen 15-3 (CA 15-3), Carcinoma Antigen 27-29 (CA 27-29) and PIK3CA. These have proved useful in monitoring the disease course in metastatic BC [99]. The clinical guidelines from the ASCO recommends using CA 15-3 and CA 27-29 as adjunctive assessments in informing treatment decisions (a ≥ 25% increase is suggested clinically significant) [100].

### 5.2. Circulating microRNAs

Several microRNAs (miRNAs) are dysregulated in BC [101]. This knowledge, coupled with the ease of isolation and their relative stability through sample processing and isolation, makes circulating miRNAs an attractive biomarker. Despite extensive research, clinically useful miRNA signatures elude oncology practice. Owing to the differences in patient selection, miRNA isolation and measurement techniques, low levels of miRNAs, concurrent diseases, effects of therapy, and insufficient studies validating its clinical utility, there has been little consensus among different miRNA panels identified so far. The future course lies in determining the most appropriate fluid for measuring miRNA (whole blood, serum or plasma), identifying the tissue of origin, standardizing the sample collection, handling and the methods of measurement, as well as the normalization of miRNA concentrations [102,103].

## 6. Metabolomics in Breast Cancer Prognosis

Equipped with the information that cancer cells display altered metabolism, a signature indicative of the presence and behavior of cancer can be generated via metabolite profiling. The information regarding the variant levels of metabolites between healthy subjects and those with cancer can be obtained by combining techniques of nuclear magnetic resonance (NMR) spectroscopy and mass spectrometry with multivariate statistical analysis. The application of metabolomics in cancer diagnostics and therapeutics is fairly new and promising, as it can provide a much-needed link between the genotype and phenotype together with some insight into oncogenesis [104]. Moreover, various studies have determined the use of this approach in predicting BC prognostic factors (ER and PR status) in tissue samples, differentiating early-stage from metastatic disease patients using serum samples, predicting BC recurrence, and predicting the response to NAC [105,106,107,108,109,110]. Additionally, metabolomics may detect micrometastasis in patients with early BC [111]. Metabolomics can also be used to search for drug metabolites in serum, in order to monitor the metabolic response to adjuvant therapy [112]. The study of the cancer metabolome is being used to identify biomarkers and potential therapeutic targets, thus also paving the way for pharmacometabolomics in cancer [104]. Akin to proteomics and transcriptomics, the assays for metabolite profiling, once established, are relatively inexpensive, rapid, and automatable [113]. Metabolomics has unbroached potential, and in the near future this new, rapidly expanding field promises to be a prime contributor to cancer diagnostics and therapeutics.

## 7. Tumor Microenvironment-Based Biomarkers: Tumor Infiltrating Lymphocytes as Prognostic and Predictive Variables in Breast Cancer

Tumor infiltrating lymphocytes (TILs) are an integral part of the tumor microenvironment and have been observed in all BC subtypes, with high counts detected in high-grade, aggressive tumors [114]. Particularly in triple-negative BC (TNBC) and HER2-positive BC, TILs display prognostic and predictive value [115,116,117]. In general, higher TIL density is associated with good prognosis and can be correlated with pathologic complete response (pCR) [115,118,119,120,121,122,123,124,125,126,127]. However, a standard method for evaluating TILs for effective integration into clinical histopathologic practice remains lacking. In 2014, an international TILs working group published a series of methodologic recommendations for evaluating TILs in BC using hematoxylin and eosin (H and E)-stained tumor sections, and recommended that TILs be reported for the stromal compartment and assessed as a continuous parameter [128]. Later, Hida et al. modified this scoring system and classified the triple-negative and HER2-positive BCs into low (<10%), intermediate and high (>50%) TIL scores [129].

Even though several studies have demonstrated the prognostic value of TIL assessment, and standardized methods are now available, the St. Gallen International Breast Cancer Conference guidelines from 2017 still do not require the inclusion of TILs in routine pathology reporting [130]. The highlights from the 2019 conference, however, do mention the potential of TILs for further improving risk stratification [131]. The clinical utility of TILs may lie not in isolation but in conjunction with other clinicopathologic variables, such as the patient age, disease stage, alterations in the genome, and other microenvironment factors [132], and their utility is likely gain prominence as immunotherapies are introduced into clinical settings. In addition, the incorporation of TIL assessment as a prognostic variable in multivariate statistical analysis is likely to further increase the role of TILs in the near future.

## 8. Neoantigens as Biomarkers of Treatment Response

Among emerging biomarkers, neoantigens display great promise in predicting responses to immunotherapy [133]. Exome sequencing along with protein mass spectrometry has led to the identification of patient-specific tumor neoantigens that result from somatic mutations in the tumor tissue. As neoantigens correlate well with the overall somatic mutation rate, as well as with the clinical response, the assays based on them can be used to measure the drug response. For example, a neoantigen landscape has been described in tumors with a strong response to CTLA-4 blockade [134].

## 9. Digital Pathology and Tissue Phenome Analysis

Pathology has always been a driver behind precision medicine, and a H and E-stained slide is the touchstone of a pathologic analysis, especially when the tissue amounts are insufficient for molecular analysis [135]. Histopathology requires large volumes of biologic tissues to be scrutinized by highly skilled pathologists, making it time-consuming, labor-intensive, and error-prone. The advances in image analysis of stained slides now enable the objective, reproducible, and automated quantification of features, including multiple co-registered features, subcellular expression patterns of biomarkers, and cellular morphometrics from digitized tissue sections. Many of these features can be captured as continuous data, and the distances between the features can also be calculated so that spatial statistics and other exhaustive computing analyses can be employed to reveal new and potentially actionable insights that could not be uncovered by the human eye alone [136,137,138,139,140].

A 2017 study by Bejnordi et al. showed the use of convolutional neural networks (CNN) in diagnosing and classifying whole-slide images from breast biopsies into three classes (normal/benign, DCIS, and invasive ductal carcinoma [IDC]). The overall classification accuracy for their system was 81.3% [141]. More recently, Sergey et al. have devised a novel machine learning based whole slide image analysis tool that can significantly predict recurrence in DCIS patients and identify those that may benefit from additional therapy [142]. In addition, Google is applying deep-content learning technology to build an automated detection algorithm that will complement the pathologist workflow. In scanning LN metastasis, their approach demonstrated substantially lower false-negative rates compared with those ascertained by pathologists [143]. Furthermore, the deep neural networks (DNNs) function by anatomizing images into pixels and sequentially combining them into features representing specific diagnostic patterns. They can also be trained to integrate follow-up studies into diagnostic algorithms, thus expanding their intelligence and enhancing the overall performance. Foreseeable is a cloud-based online DNN image analysis tool that can provide consensus on cancer diagnosis, which can be meaningful for pathologists working at small centers in receiving timely, cost-effective second opinions [144]. Although deep-learning based technologies are currently at an incipient stage, they are poised to become indispensable to personalized and precision oncology in the near future.

## 10. Scoring Centrosome Amplification: An Emerging Prognostic Marker

Centrosome amplification (CA) (i.e., an abnormal increase in the number and/or volume of centrosomes) is commonly observed in pre-invasive lesions, such as DCIS as well as invasive breast tumors. Its reputation as a BC prognostic marker strengthened in light of research that showed, high-risk BC subtypes such as TNBC patients, display higher CA (and overexpression of CA associated genes), which is associated with tumor aggressiveness and poor clinical outcomes [145]. Furthermore, CA status is connected to metastatic risk and progression free survival. A different study revealed that CA strongly correlates with higher BC grade and stage, and leads to chromosomal instability, thus driving intratumoral heterogeneity [146]. This formed the basis for a continued focus on, and development of CA as a scorable marker. Profiling CA conventionally, is technically challenging and requires advanced approaches to enhance its clinical utility.

### 10.1. Immunofluorescence-Based Three-Dimensional Image Analysis Yields a CA Score for Ductal Carcinoma In Situ Stratification

Mittal et al. have rigorously quantitated structural amplification of centrosomes in tumor samples by integrating immunofluorescence confocal microscopy with digital image analysis via an IP-protected semi-automated pipeline technology. This yields a quantifiable biomarker, the centrosome amplification score (CAS), which can stratify DCIS into low- and high-CA categories, the latter associated with a greater risk of local recurrence. The CAS may prove to be superior to the VNPI in terms of predicting the risk of local recurrence for women with DCIS of the breast [147].

### 10.2. CA20: A Transcriptomic Signature

Ogden et al. identified and validated CA20, a 20 gene CA transcriptomic signature (consisting of centrosome structural genes and genes involved in inducing CA), in breast tumors. CA20 is an independent predictor of poor survival and a high score is indicative of high chromosomal instability (CIN) and tumor aggressiveness [148]. Later, Almeida et al. in their pan-cancer computational analysis (from The Cancer Genome Atlas [TCGA] cohort) confirmed the association of CA20 with clinical and molecular features of BC as well as with CIN. In their study, CA20 correlated with poor prognosis in eight different cancers (including BC) and can aid stratification of patients with BC. In concert with immunofluorescence, CA20 can prove to be a valuable marker [149].

## 11. Prognostic Breast Cancer Staging

A major challenge for large-scale precision medicine research is in harmonizing data from different sources. This can be overcome by standardizing nomenclature and developing sophisticated metadata descriptions that enable data integration, and by enabling the portable reproducible reanalysis of datasets. In this context, two prominent BC staging systems are being discussed, the American Joint Committee on Cancer (AJCC) Tumor/Node/Metastasis (TNM) classification and the Neo-Bioscore system (University of Texas MD Anderson Cancer Center).

Recognizing the need to incorporate biologic factors (tumor grade, proliferation rate, and ER/PR/HER2 expression) and gene expression-based prognostic panels, in addition to the traditional anatomic factors into its staging system, the AJCC recently extensively revised its eighth edition of the TNM classification system, which remains the worldwide basis for BC staging. Furthermore, the AJCC Breast Expert Panel has recommended providing two BC Prognostic Stage tables; the Clinical Prognostic Stage Group (based on history, physical examination, imaging studies, and relevant biopsy results) and the Pathologic Prognostic Stage Group (for patients who have surgical resection as the initial treatment).

Furthermore, emphasizing and lending credence to the importance of tumor biology as critical for BC prognosis, researchers at the MD Anderson Cancer Center have developed the Neo-Bioscore system, a new BC staging system that builds on their previously developed CPS+EG staging system. The CPS+EG system predated the routine use of trastuzumab (Herceptin) in the neoadjuvant setting, and uses the clinical stage of cancer before NAC treatment and the pathologic stage post-NAC (CPS score), the estrogen receptor status (E), and cancer grade (G). The system generates a score that helps estimate 5-year distant metastasis-free survival post-NAC, but is unable to provide prognostic information for patients with HER2-positive disease. The Neo-Bioscore staging system overcomes this limitation by incorporating HER2 status into the score [150].

## 12. Future Perspectives

Although NGS, the backbone of multigene assays and whole-exome/genome sequencing, has vastly improved our understanding of the origin and evolution of cancer, it has generated a deluge of information and new quandaries. The clinical interpretation of the results (e.g., how to differentiate among the errors, polymorphisms, and causal mutations), as well as properly assigning pathogenicity to variants, is a work in progress. Importantly, the possibility of overinterpreting NGS testing results may lead to unnecessary medical action and add to the high levels of anxiety experienced by patients with BC and their families. Cancer gene panels do provide a middle ground by addressing the clinical questions while avoiding information overload. Additionally, NGS data yield variants of uncertain significance and incidental findings (i.e., findings unrelated to the condition under investigation) that raise ethical-legal concerns. Moreover, privacy issues are inherent when placing genomic data in the public domain and sharing genetic information, particularly regarding heritable mutations, with genetic relatives. Clearly, physician training has become crucial. Physicians need to understand the value/limitations of the data derived from these assays in the context of clinical care, and collaborate closely with genetic counselors to help their patients receive the best treatments possible.

The global profiling of an individual’s tumor at the genomic, proteomic, transcriptomic, metabolomic, and other – omic levels is technologically achievable now. A case in point is GPS Cancer, a molecular test by NantHealth that integrates whole-genome and whole-transcriptome sequencing with quantitative targeted proteomics of both normal and cancerous tissue. The presence of protein biomarkers in tumor cells at levels as low as attomoles/µg of tumor tissue can be ascertained, providing actionable insights for immunotherapy, chemotherapy, targeted therapy, hormonal therapy, and monoclonal antibody therapy. DNA, RNA, proteomic, copy number variant, and other information can now be seamlessly woven together using sophisticated big data analytics to generate probabilistic causal networks, improving the chances of identifying the perturbations that drive a tumor’s biology, or enabling a deeper segmentation of heterogeneous patient cohorts for precision medicine. However, all data are not created equal, and multi-omic analysis can be time- and cost-prohibitive. Thus, different variables might need to be accorded different weights depending on the subtype of BC and/or the stage of disease (early versus advanced) to optimize the treatment and achieve the best outcomes possible. There is also a pressing need for multivariate prognostic/predictive models that are not plagued by a lack of biomarker standardization and inter-observer variability. A useful addition to the current models would be that of a monitoring tool/biomarker (for a period > 10 years) to address the issue of late disease recurrence.

Projects like the International Cancer Genome Consortium (ICGC) and The Cancer Genome Atlas (TCGA) are working toward a shared goal of cataloging oncogenic mutations in order to further our understanding of the genetic basis of cancer, which can have a direct bearing on the diagnosis and management of the disease. The TCGA has generated exhaustive, multidimensional maps of crucial genomic alterations in 33 cancer types. The ICGC is coordinating genomics studies in tumors from 50 different cancer types and/or subtypes. The International Cancer Genome Consortium for Medicine (ICGCmed) will link the (as yet) aggregated genomics data to clinical and health information and response to therapies. The concerted efforts of these and other similar global projects are generating huge amounts of data (e.g., the TCGA dataset, accessible to the public, is currently pegged at 2.5 petabytes) that comprise RNA sequencing, proteomic, and imaging data, apart from genomic data.

To effectively employ NGS and the various biomarkers in the field of precision medicine requires complement with next-generation functional diagnostic technologies. The next-generation functional testing approach works around the limitations of traditional chemosensitivity tests by involving new cultivation methods for patient-derived tumor cells ex vivo, entailing the monitoring of live tumor states, and exposing a patient’s tumor biopsy ex vivo to drugs, thus directly revealing the cellular response to the applied agent. This does not require prior knowledge of a drug’s mechanism of action, and if the tumor cells are sensitive (usually measured by the level of tumor cell death), the drug can be clinically administered promptly [151]. Novel methods of tumor manipulation have been established that include organoids developed from cultivating single patient-derived cells, or artificial organotypic cultures developed from multiple patient-derived cells. Organoids grow in three dimensions, proving to be superior to two-dimensional cultures by more accurately mirroring the endogenous architecture of the parent tissue [152,153,154]. In addition, patient-derived xenograft (PDX) mouse models (in which biopsy material is subcutaneously or orthotopically implanted and expanded in vivo) can be generated that more faithfully recapitulate the patient’s tumor environment [155] and can be used as avatars for drug testing. Another cutting-edge development involves in situ functional diagnostics, wherein the drug effects can be directly tested with micro-dosing of solid tumors using novel devices inside the patient [156,157]. These tests require further preclinical and clinical validation to be of value for treating patients with BC, and to be incorporated into the clinical setting.

## 13. Conclusions

Cutting- edge - omics technologies, digital pathology, and new multibiomarker assays, together with clinical annotation, have granted unprecedented insights into the biology driving tumor development and the exploitable vulnerabilities, and are daily transforming the management of BC. The key lies in the effective integration of these multi-platform data and mining it for meaningful information. While the field of precision medicine in BC is making remarkable strides, it is important not to lose sight of the potential roadblocks ahead and the limitations of the current biomarkers and technologies. One such limitation is the lack of sensitive and specific biomarkers for early detection of BC. Another matter of concern is the copious amounts of genomic data being generated currently, which is only set to multiply in the future. Based on the current infrastructure, most research organizations will struggle to store and manage these data, let alone optimally analyze them. There is also the problem of underrepresentation of minority populations in most research cohorts. Detecting significantly mutated genes and all alterations involved, requires the interrogation of thousands of samples. There is thus a pressing need for increasing sample sizes for precision medicine research. Finally, increasing cohort size tackles the problem that not all genetic variants associated with disease are equally common, or equally easy to detect. As the number of sequenced genomes grows, it may be found that rare variants make important contributions to many diseases.

## Figures and Tables

**Figure 1 cancers-11-01325-f001:**
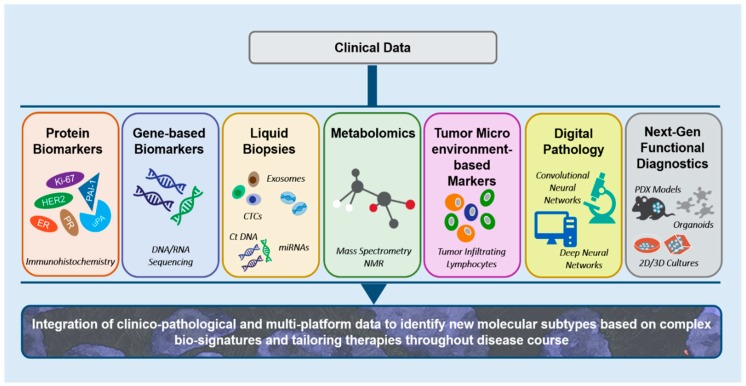
An overview of the various data generating hubs that allows integration of clinico-pathological and multi-omics data. This wealth of information can be meaningfully mined to identify new molecular subtypes based on complex multi-omics generated bio-signatures that can facilitate tailored therapies throughout the disease course in breast cancer patients.

**Figure 2 cancers-11-01325-f002:**
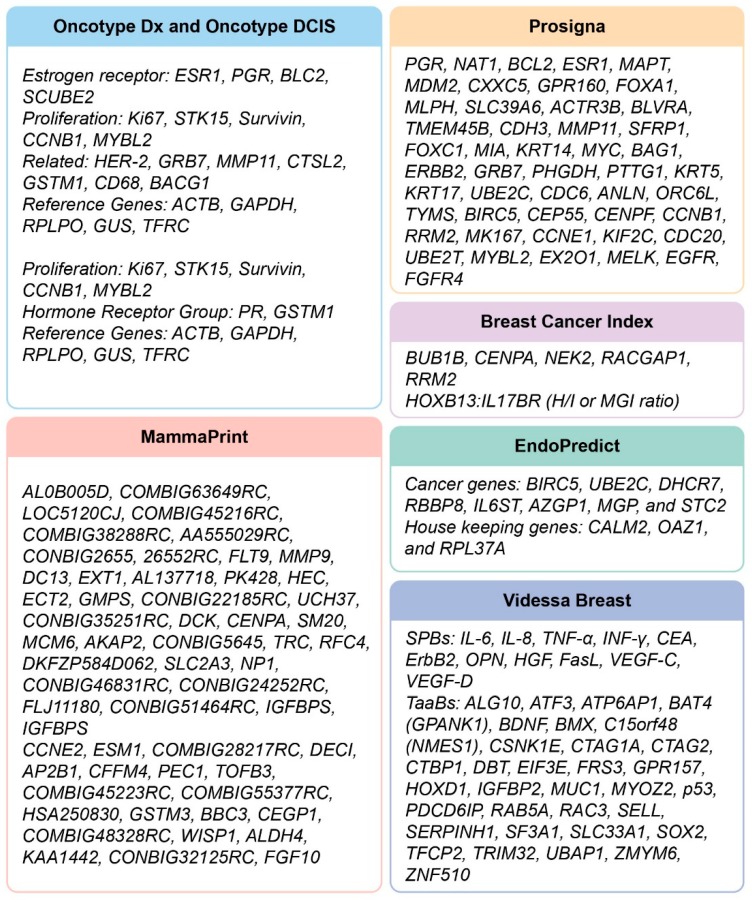
Gene/protein signatures of the prognostic/predictive multigene tests for breast cancer. SPBs [Serum Protein Biomarkers]; TaaBs [Tumor-associated autoantibodies].

**Table 1 cancers-11-01325-t001:** Prognostic/predictive multigene tests routinely used in the clinical settings for breast cancer. FFPE/FPET [Formalin-fixed Paraffin-embedded].

MGT/IHC Assay and Provider	Tissue Type, Technique, Facility	Endorsement	Clinical Indications	Prognostic/Predictive Value	Risk Groups/Stratification and Implications	Trials and Validation Studies	Comparative Advantage
**Oncotype DX** Genomic Health [20,21,22,23,24,25,26,27,28,29,30,31,32,33,34,35,36,37,38,39,40,41]	FFPE, qRT-PCR, Centralized	NCCN, ASCO, St Gallen	ER+, 0–3 node+, Stage I–II invasive, Treatment decision with tamoxifen or aromatase inhibitors	Prognostic for distant recurrence (5–10 years). Predictive for chemo and radiation sensitive in high recurrence score group. Oncotype DX DCIS is predictive of DCIS recurrence. Benefits women who have had surgery for DCIS, whether additional adjuvant treatment (radiotherapy or tamoxifen) is needed based on their risk score.	Continuous Recurrence Score (formerly triple risk stratification; intermediate score discarded on basis of TAILORx trial results): Low risk (RS 0–25; no additional benefit with chemotherapy), High Risk (RS 26–100; substantial chemotherapy benefit). Risk score for ipsilateral recurrence (invasive or DCIS); Low risk < 39, Intermediate risk 39–54, High risk ≥ 55.	TRANS ATAC, NASBP B 14/B 20, RxPONDER, TAILORx, ECOG-ACRIN, Ontario study	Considered the gold standard in MGTs with high amplification efficiency, precision and linearity.
**MammaPrint** Agendia [42,43,44,45,46,47,48,49,50,51,52]	Fresh/frozen or FFPE, Microarray, Centralized	FDA, St Gallen	Stage I–II, 0–3 node+, ER+	Prognostic for short-term distant recurrence (0–5 years). Predictive for chemoresponse in high risk group, ER+ cancer. Strong predictor of 10-year metastasis-free survival.	Binary risk classification (MP low risk or MP high risk) for recurrence without adjuvant chemotherapy. Combined with BluePrint (a molecular subtyping test) stratifies patients into four subgroups: Luminal-type/MP Low Risk; Luminal-type/MP High Risk; HER2-type and Basal-type.	TRANSBIG, MINDACT	In contrast to Oncotype DX, test was devised from patients with no hormonal (tamoxifen) or chemo-therapy and thus its robust prognostic ability. The test is endorsed for the clinical high risk group (OncotypeDx is endorsed for clinical low risk group).
**Prosigna (PAM 50)** NanoString Technologies [53,54,55,56,57,58,59,60,61]	FFPE, nCounter, Decentralized; kit compatible with other pathology labs	FDA, NCCN, ASCO, St Gallen	Stage I–III, HR+	Prognostic for 10 year recurrence in stage I–III. Prognostic and predictive for adjuvant tamoxifen.	Continuous Rate of Recurrence (ROR) score: Low risk (0–40), Intermediate risk (41–60), High risk (>61).	Trans ATAC, ABCSG8, RxPONDER	Its Prediction Analysis of Microarrays (PAM) is an almost fully automated platform technology. RNA is extracted and hybridized (by hand) from FFPE tissue in much smaller quantity than other MGTs.
**EndoPredict** Myriad Genetics [62,63,64,65]	FFPE, RT-PCR, Decentralized	ASCO, St Gallen,NCCN	Early stage, ER+,Her2−	Prognostic for early (0–5 years) and late (5–15 years) distant recurrence. Predictive for benefit from both adjuvant chemotherapy as well as which patients can safely forgo extended endocrine therapy beyond five years.	The multi-gene EP test (Figure 2) and clinical factors (nodal status and tumor size) are combined into an EPClin score stratifying patients into low- or high-risk groups: EP low-risk (<5), EP high-risk (≥5); EPclin low-risk (<3.3), EPclin high-risk (≥3.3).	GEICAM 9906, ABCSG6 and ABCSG8	EndoPredict is a second-generation, multigene prognostic test.
**Breast Cancer Index** Biotheranostics [66,67,68,69,70]	FPET, Real time RT-PCR, Centralized	ASCO, St Gallen	Stage I–III, HR+, Her2−, Node–	Predictive for adjuvant aromatase inhibitor. Predictive for hormonal therapy for 5 additional years for total of 10 years. Prognostic for late distant recurrence (post-five years).	0–10 year recurrence risk score is continuous: Low risk BCI < 5.0825, Intermediate risk BCI ≥ 5.0825 to 6.5025 and High risk BCI > 6.5025. Bimodal score informs late distant recurrence: Low risk BCI <5.0825, and High risk BCI ≥ 5.0825. BCI index for predictive utility (to direct adjuvant aromatase inhibitor treatment) is determined with the H/I ratio (Figure 2) and is just a High and Low qualification.	Trans ATAC,Stockholm trial	Outperformed both OncotypeDx and Mammostrat in its 5–10 years’ prognostic ability.

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
