# Peer review of "Panoptic View of Prognostic Models for Personalized Breast Cancer Management"

_cancers, 2019, doi:10.3390/cancers11091325_

Round 1

Reviewer 1 Report

This review covers the most research progress on the personalized breast cancer management in a broadly chronological fashion. From my point of view, this is an extensive review with valuable information and thus it merits to be published. Just, I suggest some minor modifications before publication:

Since the authors summarize so many technologies and data gleaning with the significant information they generate, then how do we proceed from here? It will be much better to give a short outlook on the develops ideas for future research. Line 47: change “has” to “have”. Line 59: delete the word “a”. Line 108: how did the authors define “More recently” since most of the reference they used in this paragraph are between 2000-2014? Consider rewriting this sentence. Line 159-168: reference?

Author Response

We are very appreciative of the reviewers’ insightful and helpful comments that have improved the readability and flow of our manuscript. Please find below our point-by-point responses to the concerns raised:

Reviewer 1

Comment 1 Since the authors summarize so many technologies and data gleaning with the significant information they generate, then how do we proceed from here? It will be much better to give a short outlook on the develops ideas for future research.

Response: We thank you for pointing this out. Throughout the manuscript, we have mentioned about the usefulness, limitations, and potential for further development with regards to the different technologies. However, we agree that a general outlook and what the future research should tackle as a whole, is missing and will be a valuable addition. Thus, we have briefly touched upon that in the Conclusion section.

Comment 2: Line 47: change has to have.

Response: The correction has been made.

Comment 3: Line 59: delete the word a.

Response: The correction has been made.

Comment 4: Line 108: how did the authors define More recently since most of the

reference they used in this paragraph are between 2000-2014? Consider

rewriting this sentence.

Response: We appreciate this observation and have rectified the mistake.

Comment 5: Line 159-168: reference?

Response: The relevant references have been added.

Reviewer 2 Report

In this manuscript, Saini et al. review the state of prognostic pathological tests that are currently used for prognosis and treatment of breast cancer. The review is a reasonable compilation of literature on this topic, which may be useful to some readers. What is lacking are the insights into the biology of breast cancer as they relate to the limited usefulness of most prognostic tests. The reader would get a very optimistic view of the omics-based approaches, rather than recognizing the difficulties in translating these into breast cancer management. In this reviewer’s opinion, the review will be more useful if it frames the challenges that need to be tackled (and how, to a degree), rather than simply supporting status quo.

The authors correctly point to the problem of tumor heterogeneity (although only in passing), they do not discuss its impact on prognosis. A large degree of poor prognosis is related to a minor subpopulation of cancer cells that can persist in quiescence under a variety of challenges, including immune attacks and therapies. The authors could briefly discuss the limits of currently used omics approaches that deal mainly with the active disease that is easily visible, and possible ways of changing this picture.

Minor point: In describing multi-gene expression based prognostic tests, the authors give an impression that these tests are routinely used. It may be good to distinguish routine research use from routine clinical management.     

Author Response

Comment 1: What is lacking are the insights into the biology of breast cancer as they relate to the limited usefulness of most prognostic tests. The reader would get a very optimistic view of the omics-based approaches, rather than recognizing the difficulties in translating these into breast cancer management. In this reviewer’s opinion, the review will be more useful if it frames the challenges that need to be tackled (and how, to a degree), rather than simply supporting status quo.

The authors correctly point to the problem of tumor heterogeneity (although only in passing), they do not discuss its impact on prognosis. A large degree of poor prognosis is related to a minor subpopulation of cancer cells that can persist in quiescence under a variety of challenges, including immune attacks and therapies. The authors could briefly discuss the limits of currently used omics approaches that deal mainly with the active disease that is easily

visible, and possible ways of changing this picture.

Response: We appreciate these invaluable points. We have addressed these concerns by adding more information in section 4.2, lines 238-248.

Comment 2: Minor point: In describing multi-gene expression based prognostic tests, the authors give an impression that these tests are routinely used. It may be good to distinguish routine research use from routine clinical management. 

Response: This is a valid point and we have clarified it in line 235-236 (Section 4.2).